# Targeting KRAS in Colorectal Cancer: A Bench to Bedside Review

**DOI:** 10.3390/ijms241512030

**Published:** 2023-07-27

**Authors:** Fernand Bteich, Mahshid Mohammadi, Terence Li, Muzaffer Ahmed Bhat, Amalia Sofianidi, Ning Wei, Chaoyuan Kuang

**Affiliations:** 1Department of Medical Oncology, Montefiore Medical Center, Bronx, NY 10467, USA; fbteich@montefiore.org; 2Department of Medical Oncology, Albert Einstein College of Medicine, Bronx, NY 10461, USA; mahshid.mohammadi@einsteinmed.edu (M.M.); terence.li@einsteinmed.edu (T.L.); muzaffer.bhat@einsteinmed.edu (M.A.B.); ning.wei@einsteinmed.edu (N.W.); 3Department of Molecular Pharmacology, Albert Einstein College of Medicine, Bronx, NY 10461, USA; 4Oncology Unit, Third Department of Internal Medicine, Sotiria General Hospital for Chest Diseases, National and Kapodistrian University of Athens, 11527 Athens, Greece; amalia.sofianidi@einsteinmed.edu

**Keywords:** colorectal cancer, KRAS, cancer therapeutics, targeted therapy

## Abstract

Colorectal cancer (CRC) is a heterogeneous disease with a myriad of alterations at the cellular and molecular levels. Kristen rat sarcoma (KRAS) mutations occur in up to 40% of CRCs and serve as both a prognostic and predictive biomarker. Oncogenic mutations in the KRAS protein affect cellular proliferation and survival, leading to tumorigenesis through RAS/MAPK pathways. Until recently, only indirect targeting of the pathway had been investigated. There are now several KRAS allele-specific inhibitors in late-phase clinical trials, and many newer agents and targeting strategies undergoing preclinical and early-phase clinical testing. The adequate treatment of KRAS-mutated CRC will inevitably involve combination therapies due to the existence of robust adaptive resistance mechanisms in these tumors. In this article, we review the most recent understanding and findings related to targeting KRAS mutations in CRC, mechanisms of resistance to KRAS inhibitors, as well as evolving treatment strategies for KRAS-mutated CRC patients.

## 1. Introduction

Colorectal cancer (CRC) is the third most common cancer and the second most common cause of cancer-related death in the US [1]. CRC is a heterogeneous malignancy, and many common as well as uncommon genetic alterations and signaling pathways are involved in its pathogenesis [2]. Similar to other cancers, mutations in specific genes acting as oncogenes or tumor suppressors contribute to CRC pathogenesis. In CRC, such mutations can be broadly categorized as sporadic, inherited, and familial [3]. The genomic and molecular heterogeneity of CRC translates into a variety of different clinical and pathological features, which, in turn, leads to different prognoses and treatment resistance patterns [2,4].

Approximately 70% of CRCs follow the adenomatous polyposis coli (APC) pathway of development and are characterized morphologically by the classic progression of normal epithelial tissue to adenoma to carcinoma [3]. This signature adenoma-to-carcinoma sequence involves a succession of alterations or “hits” affecting multiple genes. The first mutation typically occurs as a non-inherited, sporadic mutation in the tumor suppressor gene *APC*, which drives the formation of pre-malignant adenomas. Approximately 15% of these adenomas are expected to transform into carcinoma within ten years. The *APC* mutation is often followed by additional mutations in Kirsten rat sarcoma (*KRAS*), *TP53*, and, finally, *DCC* [2,5]. *KRAS* encodes a protein that regulates diverse cellular functions such as cell growth, maturation, and death [6]. Oncogenic *KRAS* point mutations lead to the permanent activation of KRAS-dependent pathways and have been identified in multiple types of cancer, most notably in pancreatic cancer, non-small cell lung cancer (NSCLC), and CRC [7,8]. It is estimated that up to 25% of all malignancies are at least partially driven by a KRAS mutation [9].

Local and locally advanced CRC is typically treated with a combination of surgery, cytotoxic chemotherapy, and, in cases of rectal cancer, with radiation therapy. However, the cornerstone of treatment for unresectable and metastatic CRC is systemic therapy [10]. Treatment typically involves multiple classes of cytotoxic chemotherapy (fluoropyrimidines such as 5-fluorouracil or capecitabine, the platinum agent oxaliplatin, and the topoisomerase inhibitor irinotecan) as well as targeted antibody therapies (anti-VEGF such as bevacizumab, and anti-EGFR such as cetuximab or panitumumab) [10,11]. KRAS-mutated CRC is notable for having poor prognosis and resistance to anti-EGFR therapies due to the KRAS-mediated constitutive activation of the MAP kinase pathway [12,13,14,15]. Furthermore, as previously stated, up to 40% of CRCs carry KRAS mutations. For all of the aforementioned reasons, novel treatments to target KRAS-mutated CRC constitute a significant area of unmet need [16,17].

In this review, we introduce the concept and characteristics of RAS mutations in CRC and aim to critically assess previous and ongoing preclinical and clinical treatment directions in the KRAS-mutated disease subset. We highlight the past and present challenges of targeting KRAS-mutated CRC. We examine the prospects for direct KRAS-targeting drugs, and the reports on inhibitors of KRAS downstream and upstream signaling pathways. Ultimately, we discuss published studies and literature on synthetic anti-KRAS compounds and evaluate whether this direction in drug development might yield alternative fruitful strategies in CRC treatment.

### KRAS Molecular Structure, Function, and Mutation

Kirsten rat sarcoma (*KRAS*) is a frequently mutated oncogene in human tumors, particularly when it pertains to CRC [15]. The KRAS protein has a molecular weight of 21 kDa and is made up of six beta strands and five alpha helices, with two major domains: the G domain and the C-terminal structural element [18,19]. The G domain, which is very well conserved, contains the switch I and switch II loops, which are responsible for GDP-GTP exchange [20]. The C-terminus is a hypervariable region containing the CAAX motif, which plays an important role in regulating KRAS localization. The CAAX motif is the target of post-translational cysteine prenylation, namely, farnesylation, which is required for the localization of the KRAS protein to the inner surface of the plasma membrane [21].

Nearly 85% of *KRAS* mutations in CRC occur in one of three main hotspots: codons 12, 13, and 61 [22,23]. The most prevalent mutations are in codon 12, accounting for approximately 65% of all *KRAS* mutant alleles. Moreover, G12D and G12V are the two most common subtypes in CRC (Figure 1) [15]. Each mutation has a distinct biological signature, with the most variability seen between the *KRAS* G13D and G12D mutation, which exhibits different treatment responses and outcomes in CRC cell lines [24]. Two other RAS isoforms have also been found to contain oncogenic mutations in CRC: Harvey and neuroblastoma rat sarcoma viral oncogenes (*HRAS* and *NRAS*, respectively) [2,25]. *KRAS* itself encodes two highly related protein isoforms, KRAS-4B and KRAS-4A, containing 188 and 189 amino acids, respectively [13,26]. These isoforms are both involved in cell proliferation, differentiation, and cell growth through the RAS/MAPK signaling pathway [27]. Generally, “KRAS” refers to the KRAS-4B isoform due to the high level of mRNA encoding KRAS-4B in cells [28]. The KRAS-4B isoform contains a phosphorylation site at serine 181 within the hypervariable region, which acts as an electrostatic farnesyl switch. The farnesyl shifting causes the translocation of KRAS-4B from the plasma membrane to the endomembrane region [29,30].

When it comes to its function, KRAS is a guanine triphosphatase (GTPase) that is typically found on the inner part of the cell membrane, anchored by a posttranslational lipid modification at its C-terminus site [31]. RAS is inactive when bound to guanosine diphosphate (GDP) and is active when bound to guanosine triphosphate (GTP). Once bound to GTP, KRAS actively transduces the signal from cell surface receptors such as Epithelial Growth Factor Receptor (EGFR) by binding to downstream effector proteins and ultimately causing cell growth and proliferation [32]. KRAS-GTP feeds into several downstream pathways, including the rapidly accelerated fibrosarcoma (RAF)-mitogen-activated protein kinase (MEK)-extracellular signal-regulated kinase (ERK) pathway and the phosphatidylinositol 3-kinase (PI3K)-protein kinase B (AKT)-mechanistic target of rapamycin (mTOR) pathway. These pathways stimulate cancer cell growth and survival, often by boosting MAPK signaling, which, in turn, increases phospho-ERK, which leads to the enhanced transcription of response genes [15,21,33]. Upon activation of PI3K by G protein-coupled receptors, the ensuing positive regulation of AKT leads to the phosphorylative activation of transcription factors such as NF-κB. KRAS-GTP binding also triggers guanine nucleotide exchange factors for the RAS-related protein Ral (RalGDS) and phospholipase Cε (PLCε) [34]. The activation of RalGDS stimulates the RAL1 binding protein and inhibits FOX transcription factors involving cell growth and proliferation. RalGDS can also promote cell survival through the JNK signaling pathway [35,36]. Due to its GTPase activity, wild-type KRAS acts as a molecular switch and turns off downstream signaling by converting its bound GTP to GDP. Point mutations at codons 12, 13, or 61 of KRAS result in blunting of the GTPase activity or diminished sensitivity to GTPase-activated proteins [14,37]. Consequently, GTP-bound KRAS protein accumulates and constitutively activates downstream effectors, resulting in abnormally increased cellular proliferation and survival [38].

In recent years, many studies have been conducted to investigate various approaches to treating CRC with KRAS mutations, a space yet to be tapped. Some KRAS-targeting molecules include synthetic anti-KRAS small molecules, KRAS downstream inhibitors, and agents that disrupt the post-translational regulation of KRAS. However, most of these attempts have been unsuccessful due to KRAS having a small GTP binding pocket with a high affinity to its native substrate GTP, preventing the binding of candidate drug molecules to KRAS. Since there is a picomolar affinity between KRAS and GTP on one hand and micromolar concentrations of GTP in the cell on the other, designing drugs to directly target the KRAS GTPase binding pocket has been challenging [31,39]. Meanwhile, it has now become well established that KRAS mutations are predictive for a lack of response to EGFR inhibitors due to the constitutive downstream activation of the MAPK pathway bypassing attempted inhibition [14,34,40,41,42,43,44]. Despite these challenges, numerous experiments have been conducted over the years and are still ongoing in an effort to validate and advance therapies that can control KRAS-mutated CRC in the clinic.

## 2. Earlier Attempts to Target KRAS-Mutated CRC

Over the last two decades, a variety of strategies have been developed to target oncogenic KRAS signaling (Figure 2). These include the development of direct inhibitors of the KRAS protein, use of RNA interference strategies, development of inhibitors that prevent the localization of RAS to the plasma membrane, and pharmacologic targeting of its downstream effectors [45]. Due to the difficulty faced in designing direct inhibitors of KRAS, many earlier attempts to target KRAS-mutated CRC focused on inhibiting signaling targets upstream and downstream of KRAS. Protein localization inhibitors, MEK inhibitors, focal adhesion kinase (FAK) inhibitors, cyclin-dependent kinase (CDK) inhibitors, and prenyl-binding protein (PDEδ) inhibitors were all studied with variable success [46,47].

As previously discussed, the farnesylation of KRAS is essential for localizing the KRAS protein to the inner surface of the plasma membrane, where it transduces the signal from EGFR to RAF. Farnesyl transferase inhibitors (FTI) were developed as early KRAS-blocking agents. Their function is to prevent the farnesylation of KRAS, blocking its migration to the cell membrane and subsequent signaling [48]. R1155777 (tipifarnib) was studied in a phase II trial that included 44 patients with stage IIIB to IV mutation unselected NSCLC. Despite pharmacodynamic data showing 83% of patients achieving inhibition of the farnesyl transferase enzyme, no objective response was achieved in the clinical trial. Seven patients had stable disease for 6 months (95% CI 6.5–10.5). The KRAS mutational status of these seven patients remains unknown [49]. Salirasib is an oral RAS farnesylcysteine mimetic that competitively blocks the membrane association of RAS proteins. It similarly failed to demonstrate any objective response in a phase II trial in advanced KRAS-mutated NSCLC. At the 10-week mark, 11 out of 33 patients had disease stability. Disease control lasted a median of 7 months. Major adverse events associated with the use of salirasib included diarrhea, nausea, and fatigue [50]. Subsequently, additional FTIs were developed and tested in KRAS-mutated solid tumors, including CRC. Lonafarnib, currently approved to reduce the risk of death due to Hutchinson–Gilford progeria syndrome, and tipifarnib failed to provide any significant benefit in multiple phase II and phase III studies, likely due to the presence of the bypass prenylation pathway via geranylgeranylation [14,51,52].

Prenyl-binding protein PDEδ is a transporter of KRAS, facilitating the diffusion of its farnesylated form in the cytosol and impacting its localization and signaling [53]. PDEδ is, therefore, a promising target to blunt oncogenic KRAS signaling. By binding to a pocket in PDEδ, the small-molecule deltarasin inhibits the KRAS–PDEδ interaction. Deltarasin has been shown to induce apoptosis and autophagy in KRAS-mutated lung cancer cell line experiments as well as decreased tumor growth in vivo [54]. In another experiment, an injection of the medication inside the peritoneum led to the regression of KRAS-mutated pancreatic adenocarcinoma xenografts compared to control animals, in a dose-dependent fashion [55]. Unfortunately, deltarasin was not very selective, which meant it was also toxic to healthy cells [56]. More selective inhibitors such as deltazinone and deltasonamide were designed. These inhibitors bind to PDEδ with high affinity, but they exhibited poor cellular potency, which limited further development [53,57].

FAK is a receptor tyrosine kinase that functions as an integrator protein, coordinating input from multiple transcellular signaling pathways involved in cell migration and adhesion. It is activated by various means including the KRAS-RHOA-FAK signaling pathway [58]. The suppression of RHOA or FAK induces cell death selectively in KRAS-mutated lung cancer cells [59]. Defactinib (VS-6063) is a FAK inhibitor that was evaluated in a phase II multi-center study with KRAS-mutated NSCLC patients who have already received one platinum doublet regimen. In total, 55 patients were treated with defactinib 400 mg orally twice daily until disease progression or unacceptable toxicity. Of these, 28% of patients met the primary endpoint, which was 12-week progression-free survival. One patient achieved a partial response. The drug was well tolerated with mostly grade 1–2 adverse events [60]. Given its relative safety and a hint at moderate single agent activity, defactinib is currently being evaluated with avutometinib (VS-6766), an inhibitor of RAS-RAF-MEK-ERK signaling, in a phase II trial dubbed RAMP-202, in KRAS-mutated NSCLC patients after the failure of prior platinum-based chemotherapy and checkpoint inhibition [61].

An alternative method that has been studied is to target *KRAS* mRNA rather than protein. AZD4785 was a nonselective anti-*KRAS* antisense oligonucleotide developed in a combined effort between AstraZeneca and Ionis Pharmaceuticals. It is a 16-mer antisense oligonucleotide that can target both wild-type and mutant *KRAS* based on the 3′ untranslated region of *KRAS* mRNA sequences outside of the mutation codons. This drug showed stability and non-target toxicity effects on CRC cells in addition to decreased proliferation in vitro exclusively in KRAS-mutated cells. In vivo experiments demonstrated AZD5785 to be a good clinical candidate as it was tolerable in monkeys and mice. This molecule inhibits MAPK and PI3K signaling pathways, while not causing MAPK reactivation in KRAS mutant models [62]. In a phase I trial (NCT03101839) with 28 participants, the intravenously administered medication was deemed safe and well tolerated. However, AZD4785 development in solid tumors has been discontinued with no published clinical results.

While not always directly translatable to CRC, the progress in targeting KRAS-mutated lung cancer can still inform on potential strategies to test in CRC. In 2011, Kim et al. reported on the biomarker-integrated approaches of targeted therapy for lung cancer elimination (BATTLE) trial, which studied different oral tyrosine kinase inhibitors (erlotinib, vandetanib, erlotinib with bexarotene, sorafenib) in pretreated NSCLC. Sorafenib achieved a disease control rate of 79% eight weeks into the treatment in a KRAS-mutated subset. Drug toxicity was, however, substantial, with 21% of treated patients requiring dose reductions and 19% discontinuation of sorafenib. In 244 patients, on different TKIs, eligible for survival analysis, the median overall survival was 8.8 months (95% CI 6.3–10.6) with a median patient follow-up of 10.3 months [63].

Bortezomib, a proteasome inhibitor (PI) commonly used to treat multiple myeloma, was also tested with mixed results in KRAS-mutated solid tumors [64]. In-human trials were planned in KRAS G12D mutant NSCLC based on preclinical data showing cellular models relying on the nuclear factor-kappa B (NF-κB) pathway and cellular death induced by inhibition of NF-κB. By downregulating the NF-κB pathway via proteasome inhibition, bortezomib induced tumor regression in KRAS G12D mutant mice models [65]. A phase II trial of subcutaneous bortezomib in patients with stage IIIB/IV or recurrent/medically inoperable NSCLC with documented KRAS G12D mutation was initiated by Riely et al. in 2013, with 16 patients enrolled. One achieved a confirmed partial response and five other patients achieved stable disease. ORR was 6% (95% CI: 0.2–30.2) and DCR was 31% (95% CI: 11.0–58.7) [66]. Peripheral neuropathy is often a limiting adverse event prompting discontinuation of PI therapy, as was the case with the single exceptional responder on this trial [66,67]. Despite this study being negative overall, similar profound responses were noted in two other patients treated with a similar approach by other providers [68]. The common denominator for all three patients was the non-smoker status and the specific histology of invasive mucinous adenocarcinoma, which often harbors G12D mutations [66].

Selumetinib and trametinib are two MEK inhibitors that were studied as monotherapy or in combination with cytotoxic chemotherapy to treat KRAS-mutated NSCLC [69]. Mouse models showed that selumetinib and docetaxel achieve synergy, tumor growth inhibition, and regression [70]. After a phase I trial showed manageable toxicity, a phase II randomized double blind trial with advanced KRAS-mutated NSCLC recruited 87 patients and randomized them to receive either docetaxel with selumetinib or docetaxel with placebo [71]. ORR, a secondary endpoint, was 37% for selumetinib. Overall survival was the primary endpoint. The trial failed to demonstrate a prolongation of overall survival with the addition of selumetinib. The median OS was 9.4 months (95% CI 6.8–13.6) in the selumetinib group and 5.2 months (3.8–NR) in the placebo group (HR 0.80, 80% CI 0.56–1.14; one-sided *p* = 0.21). However, PFS was statistically improved with a median of 5.3 months (95% CI 4.6–6.4) in the treatment group and 2.1 months in the placebo group (95% CI 1.4–3.7). Febrile neutropenia was notably more common in the treatment arm (18%) vs. no occurrence in the placebo arm [72]. A subsequent phase III trial called SELECT-1 aimed to assess the efficacy and safety of selumetinib in combination with docetaxel in patients receiving second-line treatment for KRAS-mutated NSCLC [73]. In 2017, Jänne et al. reported on 510 randomized patients—251 were treated with selumetinib + docetaxel (S+D) and 254 with placebo + docetaxel (P+D). With 447 patients available for survival analysis at the time of reporting, the median PFS was 3.9 months with S+D and 2.8 months with P+D (HR = 0.93, 95% CI 0.77–1.12; *p* = 0.44). Similarly, the median OS was not statistically different at 8.7 vs. 7.9 months, respectively (HR = 1.05, 95% CI 0.85–1.30; *p* = 0.64). ORR was 20.1% with S+D and 13.7% with P+D. A 67% grade 3 or higher toxicity occurred in the S+D arm. The development of selumetinib was abandoned after these results.

Trametinib, another MEK inhibitor, achieved a 7% ORR and 53% stable disease rate in a first study with 30 patients harboring KRAS-mutated NSCLC [74]. In a subsequent head-to-head study comparing trametinib to docetaxel after progression on platinum-doublet frontline therapy in KRAS-mutated NSCLC, progression-free survival was similar in both groups: 12 weeks in the trametinib group and 11 weeks in the docetaxel group (HR 1.14; 95% CI 0.75–1.75; *p* = 0.5197). ORR was also similar in both groups, hovering around 12%. The median overall survival was 8 months in the MEK inhibitor group and not reached in the taxane group. This trial was stopped prematurely due to futility [75]. Despite these initial setbacks in lung cancer, some scientists think there might still be a role for MEK inhibitors in the landscape of KRAS-mutated tumors, especially in combination with other chemotherapeutic agents or biologics. The ongoing work and future potential for this treatment strategy in CRC are discussed below.

In a unique study describing the resistance mechanisms of CRC against MEK inhibitors, van Shaeybroeck et al. demonstrated that MEK inhibitors cause the upregulation of c-MET/JAK/STAT3 signaling. This adaptive resistance mechanism was found to act as a bypass oncogenic signaling pathway in KRAS-mutated CRC, but not in KRAS WT disease. The authors went on to demonstrate that combination treatment with the MEK inhibitor AZD6244 and c-MET inhibitor crizotinib led to the synergistic inhibition of KRAS-mutated CRC tumor growth in both HCT116 and SW620 xenograft mouse models [76]. This study suggested that the inhibition of MEK and c-MET could be a rational treatment combination in KRAS-mutated CRC. The MEK and MET Inhibition in Colorectal Cancer trial (MErCuRIC1) was a phase I clinical trial aiming to test the combination of crizotinib with a MEK inhibitor, either binimetinib or PD-0325901, in CRC patients with either a KRAS mutation or aberrant c-MET activation. While full results have not yet been published, the dose expansion treated 36 patients. Of these, 30 patients were evaluable for response, with a 0% ORR, 23.3% DCR, a PFS of 1.81 months, and OS of 5.42 months (NCT02510001).

Following the MEK inhibitor rationale for KRAS-driven tumors, a more recent study demonstrated that the treatment of HCT116 and SW480 CRC cell lines, both of which are KRAS-mutated, with the MEK inhibitor AZD6244 caused the upregulation of multiple oncogenes including EGFR, MET, FAK, STAT3, and AKT. This was also found to be due to bypass RTK signaling through the growth factor receptor-bound protein 7 (GRB7). The investigators demonstrated that combination treatment with AZD6244 as well as the GRB7 axis was able to synergistically suppress KRAS-mutated CRC tumor growth in the HCT116 xenograft model [77]. This study further supports the concept that single pathway inhibitors to target KRAS-driven signaling will typically lead to an acute adaptive resistance through bypass signaling pathways, and that combination therapies will most likely be needed to adequately treat these tumors.

Cyclin-dependent kinases (CDKs) are protein kinases involved in cellular proliferation and transcription, through the regulation of cell-cycle checkpoints and transcriptional events in response to extracellular and intracellular signals [78]. Based on their functions, CDKs may be divided into two main sub-groups: cell cycle CDKs (CDK1, CDK2, CDK4, CDK6) and transcriptional CDKs (principally CDK7, CDK8, CDK9) [79]. CDK4/6 inhibitors have been shown to be active agents in early and metastatic hormone-receptor-positive breast cancer, in combination with endocrine therapy such as fulvestrant or letrozole. Three agents, namely, abemaciclib, palbociclib, and ribociclib, have all improved progression-free survival compared to placebo and led to drug approvals by the FDA and EMA [80]. Preclinical data suggest that CDK function is critical for KRAS-driven tumorigenesis and that the inhibition of CDK in KRAS-mutated NSCLC leads to potent synthetic lethality [81]. In CRC, KRAS-mutated tumors have been shown to be enriched for cell cycle and mitotic processes, consistent with a dependency on CDK4/6 [82]. Based on these findings, combination therapy of CDK4/6 inhibitor palbociclib and MEK inhibitor PD0325901 was tested on KRAS-mutated CRC cells, BRAF-mutant CRC cells, normal colon epithelial cells, and CRC xenograft models to evaluate their efficacy and toxicity. This study ultimately showed that the co-inhibition of CDK4/6 and MEK caused a therapeutic response in both KRAS-mutated and BRAF-mutated CRC while demonstrating limited activity against KRAS WT and normal colon cells [82].

The concept of dual inhibition of CDK4/6 and MEK has also been studied clinically. Abemaciclib was tested against erlotinib in patients with stage IV NSCLC with a KRAS mutation progressing on platinum-based therapy in the JUNIPER randomized controlled trial. Here, 453 patients were randomized 1:1 to abemaciclib or erlotinib. The median OS was 7.4 months (95% CI: 6.5, 8.8) with abemaciclib and 7.8 months (95% CI: 6.4, 9.5) with erlotinib (HR = 0.968 (95% CI: 0.768, 1.219); *p* = 0.77). ORR was 8.9% and 2.7% (*p* = 0.01), respectively. There was a mild increase in PFS from 1.9 to 3.6 months (*p* < 0.000001) warranting the further investigation of CDK4/6 inhibition in this subgroup of tumors [83]. In the realm of CRC, combined treatment with the CDK4/6 inhibitor palbociclib and MEK inhibitor binimetinib was recently shown to synergistically suppress tumor growth across a panel of RAS-mutated CRC patient-derived xenograft models [84]. However, a phase II clinical trial of binimetinib combined with palbociclib did not result in improved survival for chemotherapy-refractory CRC patients [85]. Although this study was not positive, work is ongoing to identify the biomarkers of sensitivity to this combination.

Finally, drug repurposing is a promising strategy for identifying new KRAS inhibitors. This approach uses machine learning and artificial intelligence to screen large datasets of approved drugs for potential activity against KRAS. One of the advantages of this method, in comparison with de novo drug development, is the use of compounds having already passed preclinical and in-human safety testing, thus allowing new clinical studies to be rapidly designed and conducted in an attempt to identify new prospective KRAS inhibitors initially approved for other targets and diseases [86]. Using a screening method with organoids, Mertens et al. demonstrated the activity of vinorelbine in combination with EGFR/MEK inhibitors in RAS-mutated CRC [87]. This work led to the ongoing phase I/II RASTRIC trial, which is evaluating the efficacy of the triple combination of binimetinib, lapatinib, and vinorelbine in metastatic CRC (EudraCT 2019-004987-23). Similarly, Srisongkram et al. used computational approaches to identify afatinib, neratinib, and zanubrutinib as potential KRAS G12C inhibitors. These agents were selected after multiple steps and the application of a thorough algorithm [88]. Despite its limitations, drug repurposing, if performed correctly, can help bring new products to market more quickly than traditional drug discovery. It can also help transfer these strategies more efficiently to other tumor types driven by KRAS aberrations, such as NSCLC and pancreatic ductal adenocarcinoma.

## 3. Direct KRAS Targeting in CRC

It was not until 2013 that a major breakthrough in KRAS drug development occurred. For the first time in decades, a small pocket in the mutant KRAS G12C protein beneath the effector binding switch-II region was identified and characterized by the Shokat Lab [89]. Subsequently, Ostrem et al. reported the development of a covalent inhibitor binding to KRAS G12C, disrupting both switch-I and switch-II, changing the natural preference for GTP to GDP and impairing binding to RAF [90]. These experiments were the first evidence of KRAS “druggability”, long thought to be the “holy grail” of targeted therapies. This approach also demonstrated that direct KRAS inhibition can be achieved using a mutant allele-specific agent, sparing other KRAS mutants and normal cells with wild-type KRAS alleles. A nonselective approach to KRAS targeting could theoretically result in higher toxicity given the ubiquitous distribution of the KRAS protein.

Small-molecule inhibitors attempted to directly inhibit the G12C form of KRAS in a conformation-specific manner [91]. KRAS G12C is different from other KRAS alleles like G12D and G12V because it can continue to interact with its downstream effectors through active cycling between a GDP-bound and a GTP-bound state [31]. Targeting cysteine residues on KRAS G12C can lock it into an inactive conformation, preventing its reactivation by nucleotide exchange, halting the aforementioned cycling phenomenon and disrupting downstream signals [91]. After the successful demonstration of the initial proof of concept, chemical optimization followed. ARS-853 is a selective, covalent KRAS G12C inhibitor and at the time of its characterization, it was the first direct KRAS inhibitor with cellular potency in the range of a drug candidate [92]. However, ARS-853 suffered from poor bioavailability [93]. The first direct KRAS G12C inhibitor to enter clinical trials was Amgen’s sotorasib, formerly known as AMG510. Mirati Pharmaceutics discovered adagrasib, formerly known as MRTX849, as another covalent and potent KRAS G12C inhibitor. Other promising KRAS G12C allele-specific inhibitors with potential therapeutic effects include ARS-1620, SML-8–73–1, and ASP2453 [92,93,94,95,96,97,98]. 

### 3.1. Sotorasib

Although AZD4785 failed to progress to late-stage clinical trials, the advance in 2013 showing that the switch-II pocket (S-IIP) is potentially targetable when KRAS proteins have the G12C substitution has led to the advent of new drugs, triggering an avalanche of small-molecule inhibitors directed against allele G12C of KRAS [90]. Sotorasib was a first-in-class direct selective KRAS inhibitor developed by Amgen [99]. It hones down on the G12C mutant form of the protein, and was demonstrated in preclinical studies to induce the regression of KRAS G12C tumors and to synergize with chemotherapy and other targeted agents, especially the MEK inhibitor trametinib [100]. In addition, sotorasib was found to induce a proinflammatory milieu alone or in tandem with immune checkpoint inhibitors blocking PD-1, prompting further in-human studies of such combinations [100]. In 2019, Fakih et al. reported the results of a phase I trial evaluating the safety, tolerability, and efficacy of sotorasib in adults with locally advanced or metastatic KRAS G12C mutant solid tumors. Twenty-two patients were enrolled and treated with three different doses of the agent; 6 patients had NSCLC and 15 patients had CRC. The therapy was well tolerated with no reported drug limiting toxicities. Five patients reported grade 2 or lower treatment-related AEs. Amongst nine patients evaluable for response at the first data cut-off, one patient with NSCLC achieved a partial response and six patients, with either NSCLC or CRC, achieved stable disease [101].

A subsequent phase I from the CodeBreaK 100 program was conducted in 129 patients with heavily pretreated advanced solid tumors with a KRAS G12C mutation. Dose escalation was first completed followed by expansion in NSCLC and CRC cohorts. Here, 73 patients (56.6%) had treatment-related adverse events, and 15 patients (11.6%) had grade 3 or 4 events. The most common adverse reactions were diarrhea, musculoskeletal pain, nausea, fatigue, hepatotoxicity, and cough. The confirmed ORR was 32.2% in the NSCLC subgroup and the DCR was 88.1%. The median progression-free survival (PFS) in this group was 6.3 months. In the CRC cohort, 7.1% of patients had a confirmed objective response, 73.8% had some form of disease control, and the median PFS was 4 months. In addition to these types of cancers, there were reports of responses in pancreatic cancer, uterine cancer, and melanoma [102].

A phase II trial under the umbrella of CodeBreaK 100 evaluated 124 patients with KRAS G12C-mutant advanced NSCLC who had progressed on prior platinum-based chemotherapy and programmed death 1 (PD-1) or programmed death ligand 1 (PD-L1) therapy, and who had measurable disease. ORR was 37.1% with four (3.2%) complete responses, and a DCR of 80.6%. The median PFS was 6.8 months and median OS was 12.5 months. Grade ≥3 treatment-related adverse events (TRAEs) were reported in 26 (20.6%) patients and mainly comprised diarrhea and drug-induced liver injury [103]. These data led to the accelerated approval of sotorasib for advanced KRAS G12C-mutant NSCLC based on the objective response rate (ORR) and duration of response (DoR). The label dosing of sotorasib is 960 mg once daily with multiple dose level reductions available if toxicity arises [99].

A confirmatory phase III trial, CodeBreaK 200, was launched in 148 centers in 22 countries to evaluate sotorasib versus standard of care docetaxel in second-line KRAS G12C-mutant NSCLC with prior exposure to a platinum agent and an immune checkpoint inhibitor. In this study, 345 patients were randomized 1:1 to oral sotorasib starting at a dose of 960 mg daily or to docetaxel at 75 mg/m^2^ IV every 3 weeks. The crossover from docetaxel to sotorasib was allowed after confirmed radiologic progression. The trial’s primary endpoint was progression-free survival. There was a statistically significant difference (*p* = 0·0017) in favor of sotorasib with a median progression-free survival of 5.6 months (95% CI 4.3–7.8) compared to 4.5 months (95% CI 3.0–5.7). There were fewer grade 3 or worse treatment-emergent adverse events (TEAEs) in the treatment arm (11%) compared to the control arm (23%). Despite the slight improvement in PFS, the median overall survival was no different between the groups, reaching 10.6 months (95% CI 8.9–14.0) in the sotorasib group vs. 11.3 months (95% CI 9.0–14.9) in the docetaxel group (HR = 1.01, 95% CI = 0.77–1.33, *p* = 0.53) [104].

Similarly, sotorasib was evaluated in an international single-arm phase II trial in patients with locally advanced unresectable or metastatic CRC with a KRAS G12C mutation who had progressed on prior fluoropyrimidine, oxaliplatin, and irinotecan. In total, 62 patients were enrolled. In line with phase 1 data in CRC, ORR was 9.7%, DCR was 82.3%, and the median PFS was 4 months. Only seven (12%) patients developed a grade 3 or 4 TEAE [105]. Unlike in NSCLC, single-agent activity seems to be more limited in KRAS-mutated CRC, lending credence to the hypothesis that other mutations or pathways could be bypassing direct KRAS inhibition and driving tumor growth [106].

CodeBreaK 101 is a phase Ib/II that studied sotorasib monotherapy or in combination with other chemotherapy or targeted agents in various advanced solid tumors. In late 2022, Kuboki et al. revealed that one of the phase Ib arms enrolled 40 patients to sotorasib 960 mg orally daily and panitumumab 6 mg/kg IV every 2 weeks. The confirmed ORR was 30% and the disease control rate was 90%. Grade 3 TEAEs thought to be related to sotorasib occurred in six (15%) patients. This study was the first in-human evidence to show synergy between the agents, with a threefold increase in response rates compared to sotorasib monotherapy [107]. All of these observations culminated in the initiation of CodeBreaK 300, a phase III randomized, open-label, active-controlled comparison of sotorasib and panitumumab versus the investigator’s choice of therapy (trifluridine/tipiracil or regorafenib) in patients with previously treated metastatic CRC. This study is currently ongoing and, if positive, may lead to a paradigm change in the third-line setting in KRAS-mutated CRC (NCT05198934). 

### 3.2. Adagrasib

Adagrasib was the second approved KRAS inhibitor to follow in the footsteps of sotorasib [108]. On paper, it has the theoretical advantages of having a long half-life (23 h), dose-dependent pharmacokinetics, and central nervous system (CNS) penetration [109]. It was approved in December 2022 for adult patients with KRAS G12C-mutated locally advanced or metastatic NSCLC based on positive results from the NSCLC cohort from the registrational study KRYSTAL-1 [108,110].

KRYSTAL-1 was a multicohort phase I/II study evaluating adagrasib as monotherapy or in combination regimens in patients with advanced solid tumors harboring a KRAS G12C mutation. Adagrasib 600 mg orally twice daily or adagrasib in combination with cetuximab (400 mg/m^2^ followed by 250 mg/m^2^ every week or 500 mg/m^2^ every 2 weeks) were regimens evaluated in cohorts with pretreated metastatic KRAS G12C-mutant colon cancer. As of June 2022, 44 patients received adagrasib alone and 32 patients received the combination. In the monotherapy cohort where 43 patients were evaluable for efficacy, the ORR was 19% and DCR was 86%. The median DoR was 4.3 months (95% CI 2.3–8.3) and median PFS was 5.6 months (95% CI 4.1–8.3). In the adagrasib/cetuximab cohort, amongst 28 evaluable patients, the ORR was 46% and DCR was 100%. The median DoR was 7.6 months (95% CI 5.7–NE) and median PFS was 6.9 months (95% CI 5.4–8.1). Grade 3 or higher TEAEs occurred in 34% of patients on adagrasib and in 16% of patients on dual therapy [111].

KRYSTAL-10 is an ongoing phase 3 trial currently investigating adagrasib with cetuximab versus standard fluoropyrimidine-based chemotherapy in second-line metastatic KRAS G12C-mutated colon cancer (NCT04793958). Another ongoing trial of interest includes NCT05722327, which is a phase I effort aiming to study the combination of adagrasib, cetuximab, and irinotecan in KRAS G12C CRC treated with two previous lines of therapy (NCT05722327).

### 3.3. Comparison between Sotorasib and Adagrasib

Both sotorasib and adagrasib target the same P2 pocket, trapping the oncoprotein in an inactive GDP-bound state. Despite structural differences such as a longer half-life and better CNS penetration for adagrasib, the response rates and survival outcomes are similar in NSCLC, keeping in mind the limitations of cross-trial comparisons. In KRYSTAL-1, adagrasib seemed to cause more G3-G4 adverse events, mostly fatigue and aminotransferase elevation, whereas diarrhea, nausea, and liver dysfunction were more common with sotorasib. CodeBreaK 100 excluded patients with active brain metastasis, while KRYSTAL-1 allowed such patients and showed an intracranial confirmed objective response rate of 33.3% with a median duration of intracranial response of 11.2 months in a post hoc analysis [111]. In the absence of head-to-head trials, more research is required to study the potential differences in toxicity profile and intracranial activity between the agents [112].

### 3.4. Mechanisms of Resistance to Sotorasib and Adagrasib

Progression after an initial response to KRAS inhibitors is inevitable. Intratumoral heterogeneity seems to play a large role in this resistance phenomenon [113]. There are mechanisms of rapid adaptation to continuous KRAS inhibition. After an initial phase of inhibition, cell lines show a re-accumulation of active KRAS, bound to GTP [91]. Li et al. evaluated patients with NSCLC and CRC upon progression on sotorasib, and 28% of NSCLC patients and 73% of CRC patients developed at least one acquired genomic abnormality at the time of progression. The most common alterations involved receptor tyrosine kinase pathways such as EGFR, MET, FGFR2, FGFR1, MYC, and ROS1. They occurred in 24% of NSCLCs and 27% of CRCs, respectively. Secondary RAS alterations (single-nucleotide variants, copy number variant amplifications) were the second most common mechanism of resistance in CRCs [114]. Cell cycle alterations, MAPK pathway activation, epithelial–mesenchymal transition and tumor microenvironment remodeling are all thought to be part of drug escape strategies [115,116].

Zhao et al. studied pre- and post-treatment specimens from 43 patients treated with sotorasib. They identified the emergence of multiple subclonal events during G12C inhibition. Acquired oncogenic KRAS, NRAS, and BRAF mutations were identified, suggesting a role for ERK signaling co-blockade [117].

## 4. Future Directions

Investigators are working to develop new ways to inhibit KRAS. They are looking to target isoforms other than G12C, developing more potent and selective second-generation inhibitors, and combining KRAS inhibitors with other therapies such as MAPK pathway inhibitors, alternate oncogene pathway inhibitors, and immune checkpoint inhibitors. The goal of such advances is to broaden and deepen tumor response, prolong survival, and overcome resistance to monotherapy [118].

### 4.1. Combination with EGFR Inhibitors

Inhibitors of the MAPK signaling pathway may provide an alternative solution for future CRC treatments to overcome drug resistance [119,120]. Studies have demonstrated that the mutant KRAS G12C in CRCs are dependent on upstream receptor tyrosine kinase signaling (RTK), particularly EGFR. Biochemical approaches have shown that under the treatment pressure from KRAS G12C inhibitors, CRC will upregulate EGFR, leading to adaptive resistance to KRAS G12C monotherapy. The rapid rebound of RTKs, especially EGFR, was responsible for CRC resistance to KRAS G12C inhibitors [121]. Combination therapy targeting both EGFR and KRAS G12C was remarkably effective in CRC cells. The combination of cetuximab and sotorasib inhibited the activation of the EGFR-driven MAPK pathway in these cells, thus sustaining downstream target inhibition, considerably increasing KRAS G12C inhibition and leading to tumor regression [122]. These studies have led to the currently ongoing clinical trials investigating KRAS G12C inhibitors combined with EGFR inhibitors, as previously discussed.

### 4.2. Combination with MEK Inhibitors

Targeting MEK in addition to KRAS is a logical way to further suppress this pathway. Preclinical experiments have demonstrated synergy between MEK inhibitors and sotorasib in cell lines and patient-derived xenografts [100]. In a separate study, the combination of sotorasib or ARS-1620 (a G12C inhibitor with similar ligand structure), with a variety of RTK pathway inhibitors such as SHP099, erlotinib, afatinib, crizotinib, or BGJ398, all showed synergistic effects against KRAS G12C CRC cell lines and xenografts [123]. This work highlights the importance of a combinatorial treatment strategy targeting multiple potential RTK feedback mechanisms, such as with MEK inhibitors, for maximal benefit.

Despite not showing significant response rates or survival outcomes compared to docetaxel in the second-line treatment of NSCLC, trametinib was evaluated in association with sotorasib in KRAS G12C-mutated NSCLC and CRC [75]. In 2021, 18 patients with CRC had been treated with sotorasib 960 mg daily in combination with trametinib at 1 mg or 2 mg daily. Some patients were previously treated with sotorasib and had evidence of progression prior to enrolment. In seven patients previously exposed to a KRAS G12C inhibitor, there was one partial response with an ORR of 14.3%, while sotorasib-naïve patients exhibited an ORR of 9.1%. DCR was 85.7% in pre-exposed patients and 81.8% in first-time patients. The combination seems to be toxic, with 34.1% of 41 patients treated in both NSCLC and CRC cohorts reporting a grade 3 or higher adverse event [124]. Triplets are also being investigated in CodeBreaK 101 with the combination of trametinib, sotorasib, and panitumumab, hoping to achieve even deeper suppression of the co-activated MAPK pathway in KRAS-mutated CRC [125].

Avutometinib, also known as VS-6766, is an RAF/MEK clamp being developed by Verastem in combination with sotorasib and/or an FAK inhibitor, defactinib, in KRAS-mutated malignancies. Preclinical studies revealed tumor regression in all mice harboring G12C-mutant NSCLC with a triple combination, which aims to achieve vertical inhibition of RAS, RAF, and MEK [126]. The first in-human studies are currently recruiting patients. The efficacy of VS-6766 in combination with direct KRAS inhibitors and/or EGFR-inhibiting antibodies in patients with KRAS-mutated CRC warrants further investigation.

### 4.3. Combination with SOS1 Inhibitors

The son of sevenless homologue 1 (SOS1) protein is universally expressed in cells and plays an important role in the RAS signaling pathway. In response to upstream stimuli, SOS1 interacts with KRAS and promotes guanine nucleotide exchange and the activation of downstream signaling pathways [127]. Interfering with the KRAS-SOS1 interaction would hinder the activation of KRAS by preventing the conversion of GDP to GTP. SOS1-KRAS inhibitors exhibit activity on a broad spectrum of KRAS alleles, including all major G12D/V/C and G13D oncoproteins [128].

BI-3406 is an SOS1-panKRAS inhibitor developed by Boehringer Ingelheim. In a CRC SW837 cell line-derived xenograft with acquired resistance to adagrasib, Thatikonda et al. showed that addition of BI-3406 to adagrasib overcame induced resistance to KRAS inhibitor monotherapy [129]. Based on these data, BI-1701963 is an SOS1 inhibitor being investigated in combination with adagrasib in the KRYSTAL-14 program (NCT04975256) and with sotorasib in CodeBreaK 101 (NCT04185883).

### 4.4. Combination with SHP2 Inhibitors

SHP2 or PTPN11 is a pleiotropic non-receptor protein tyrosine phosphatase serving as a hub connecting intracellular oncogenic pathways such as JAK/STAT, PI3K/AKT/mTOR, and RAS/RAF/MAPK to drive cellular growth [130], and helping transduce PD-1 signaling [131]. SHP2 inhibition is strongly synergistic with MEK or ERK inhibition in KRAS-mutated cancers [132]. Allosteric inhibitors that selectively bind the non-catalytic site of SHP2 have recently been developed after a decade of failed attempts to target the SHP2 catalytic site [133]. In a preclinical study, the oral drug D-1553 (garsorasib) designed by InventisBio has been assessed in CRC cell lines and patient-derived xenografts in monotherapy form or in combination with an MEK inhibitor, SHP2 inhibitor, or chemotherapeutic compounds. A reduction in tumor growth and proliferation has been observed in CRC cells as well as xenografts [134]. Additionally, it is undergoing clinical investigation in ongoing phase I/II clinical trials (NCT04585035).

Preclinical data combining SHP2 and ERK2 inhibition in KRAS-mutated pancreatic adenocarcinoma showed synergy and the triggering of apoptosis in vitro. The combination is tolerated and promotes tumor regression in multiple in vivo models [135]. Similar to activating KRAS mutations, the amplification of wild-type KRAS seems to constitute an alternative means of activating this oncoprotein in cancer. Wong et al. demonstrated that a combination of SHP2 and MEK inhibition led to a durable inhibition of the pathway in wild-type KRAS-amplified gastric cancer [136].

One of the most important ongoing first-in-human trials is the SHERPA trial, a combination therapy of SHP2 inhibitor RMC-4630 and ERK inhibitor LY3214996 in metastatic KRAS-mutated cancers [135]. Alternatively, TNO155 is a promising SHP2 inhibitor being studied in a broad clinical combination strategy to blanket the MAPK pathway. One important combination is with direct KRAS inhibitors such as in G12C-mutated cancers. KontRASt-01 is a phase Ib/II evaluating JDQ443, a new KRAS G12C inhibitor, in combination with TNO155 in advanced solid tumors including CRC. It is currently recruiting patients with an estimated completion in 2027 (NCT04699188).

### 4.5. Newer G12C Inhibitors

Now that the KRAS code has been cracked, and with sotorasib and adagrasib setting the stage for future drug development, we are witnessing a large wave of next-generation KRAS G12C inhibitors in the pipelines. Notably, Eli Lilly and Roche have particularly interesting drugs with unique theoretical advantages. LY3537982 is a new G12C direct inhibitor (G12Ci) developed by Loxo Oncology, a subdivision of Eli Lilly. It is touted to have a cleaner safety profile compared to sotorasib and adagrasib. In the phase I trial, 84 patients with KRAS G12C-mutated advanced cancers received LY3537982 monotherapy. There was preliminary efficacy across all dose levels in multiple tumor types. LY3537982 led to a 10% ORR and a 90% DCR in 20 CRC patients enrolled in the trial. These numbers were 38 and 88%, respectively, in KRAS G12Ci-naïve NSCLC. Interestingly, LY3537982 showed activity in NSCLC patients previously treated with a G12Ci, with an ORR of 7% (1/14) and a DCR of 64% (9/14). In cohort C2 of the trial, LY3537982 was combined with cetuximab in CRC. The doublet achieved an ORR of 45% (5/11) with five partial responses, six stable diseases, and a DCR of 100% [137].

LY3537982 is believed to be 10 times more potent than first-generation KRAS G12C inhibitors [138]. It can therefore be administered at lower doses, with adequate inhibition of the target and less off-target effect. No high-grade drug-related hepatotoxicity was observed in the escalation phase of the trial and no dose-limiting toxicities were observed at any dose level of the drug [137]. The further evaluation of LY3537982 in combination with ICIs and in earlier lines of therapy is warranted given the encouraging adverse effect profile.

Divarasib, or GDC-6036, is a highly potent irreversible covalent inhibitor of the KRAS G12C mutant protein, developed by Roche. Given solid preclinical data, phase I dose escalation and expansion in KRAS G12C mutant cancers started in 2020 [139,140]. The agent is exhibiting similar response rates with lesser toxicity than first-generation inhibitors in their early studies [141]. It is expected that the more potent and selective inhibitors will be more likely to advance to market in the saturated KRAS G12C inhibitor space.

### 4.6. KRAS G12D Targeting

While KRAS G12C was the first isoform to be directly targeted in CRC, G12D constitutes the most common type of KRAS mutation in CRC, affecting approximately one third of KRAS-mutated CRC patients (Figure 1) [142]. At first sight, G12D seemed to be more difficult to target than other types of KRAS mutations because it lacks a reactive cysteine residue and has a faster GTP hydrolysis rate [31]. Nevertheless, in 2022, Hallin et al. reported on a compound, MRTX1133, a potent selective noncovalent KRAS G12D inhibitor, showing high-affinity interaction with GDP-bound KRAS G12D. MRTX1133 has been shown to inhibit KRAS signaling in vitro and to cause tumor regression greater than 30% in PDX models [143]. In another experiment, Tajiknia et al. studied MRTX1133 in combination with 5-fluorouracil in pancreatic cancer and CRC cell lines. There was evidence of synergy as well as activity beyond the G12D isoform, particularly in G12V-mutated cell lines. Moreover, the combination of drugs led to a reduction in IL8/CXCL8 and TNF-alpha levels in addition to an increase in pro-inflammatory cytokine IL-18/IL-1F4, suggesting a role played via immune environment modulation [144]. MRTX1133 monotherapy is currently being evaluated in a first-in-human phase 1/2, open-label, multicenter study in patients with advanced solid tumors harboring a KRAS G12D mutation (NCT05737706). Finally, preclinical data suggest that combining MRTX1133 with EGFR or PI3K/AKT/mTOR pathway inhibitors may be synergistic in treating cancer. Other potential combinations worthy of further investigation include cytotoxic chemotherapy with a direct KRAS inhibitor and immune checkpoint inhibitors with a G12D inhibitor [145].

### 4.7. Tri-Complex and panKRAS Inhibitors

Sotorasib and adagrasib are covalent inhibitors that bind to the inactive, GDP-bound form of the KRAS G12C oncoprotein and are therefore called KRAS-G12C(OFF) inhibitors. In theory, an inhibitor that directly targets the active GTP-bound form of G12C, also known as a KRAS-G12C(ON) inhibitor, can prevent effector interaction with active KRAS and is predicted to withstand adaptive pressures that limit the effect of inactive state-selective compounds. ON inhibitors form a ternary complex with chaperone cyclophilin A, thus overcoming resistance driven by enhanced upstream signaling. They are sometimes referred to as a KRAS(G12C)-GTP:CYPA inhibitor [146,147].

RMC-6291 is one such drug. By bridging the active GTP-bound form of KRAS to cyclophilin A, it forms an inactive tricomplex preventing RAS interactions with downstream signaling partners [148]. In September 2022, Revolution Medicines started a phase I/Ib trial studying the dose escalation and expansion of RMC-6291 in patients with advanced solid tumors harboring the KRAS G12C mutation (NCT05462717). In a similar fashion, RMC-6236 is a non-selective RAS(ON) inhibitor that has demonstrated tumor shrinking across various KRAS genotypes, including G12D, G12V, and G12R mutations [149]. The first patient was dosed in the phase I/Ib monotherapy clinical trial in the middle of 2022 (NCT05379985).

The pan-RAS inhibitor MCI-062 has been shown to exhibit anti-tumor and anti-KRAS activity in multiple mouse models of KRAS-mutated CRC. MCl-062 was effective against HCT116 human colon cancer cells, which harbor KRAS G13D, and was ineffective against HT29 cells harboring BRAF V600E [150]. This study also performed in vivo testing of MCI-062 on a syngeneic mouse model of KRAS-driven CRC (CT-26) and demonstrated that MCI-062 acted on its target, reduced GTP-RAS, suppressed the activation of the MAPK signaling pathway, and inhibited tumor growth. MCI-062 also suppressed PD-L1 expression and activated anti-tumor immunity [150]. Further investigation of this drug in clinical trials is awaited.

Kim et al. recently published the characterization and preclinical anti-tumor activity of the novel panKRAS inhibitors BI-2865 and BI-2493 [151]. These molecules are non-covalent inhibitors of the inactive GDP-bound KRAS state. These two drugs are structural homologs of one another, with BI-2493 being optimized for in vivo administration. These panKRAS inhibitors were demonstrated to inhibit tumor growth and downstream MAPK signaling in an extensive panel of KRAS-mutated and KRAS-activated cancer cell lines. While there was some variability in the degree of inhibition of MAPK signaling based on the specific KRAS mutant allele, the investigators found that KRAS mutant tumors were generally more sensitive to inhibition by the drugs than non-KRAS mutant tumors. The clinical investigation of this family of molecules is also eagerly awaited.

### 4.8. Other Emerging Therapies

Cellular therapies, including Chimeric Antigen Receptor (CAR) T-cell therapy and T-cell receptor (TCR)-based adoptive T-cell therapy, as well as vaccine therapies, are promising new mechanisms to target KRAS [152]. In 2016, Tran et al. reported on a patient with metastatic CRC harboring a KRAS G12D mutation, and responding to TCR therapy recognizing peptide fragments of KRAS G12D presented by HLA molecules [153]. This was proof of concept that tailored immunotherapy approaches are feasible, effective, and can lead to long-lasting disease control when they work. In 2022, Leidner et al. also reported on the success of a neoantigen T-Cell receptor gene therapy in pancreatic cancer with a KRAS G12D mutation. TCRs were HLA-C*08:02-restricted. One infusion of cells led to a regression of visceral metastases lasting at least 6 months [154]. These experiments provide hope for KRAS-mutated cancers. However, several challenges remain before TCR therapy can be widely adopted, including widespread institutional feasibility, cost, and required patient preconditioning. HLA restriction is also a hurdle that needs to be overcome to make therapy more accessible to the general patient population [155].

Vaccines are another approach developed to target KRAS and are being studied in both the adjuvant and metastatic settings. Early vaccines had minimal clinical activity and generated mostly a CD4 T-cell response, which is not dramatically effective against mutated KRAS [156,157]. Dendritic cell vaccines are currently being developed with the hope of generating a stronger immune response directed at KRAS [158].

A recently published study using a mouse embryonic fibroblast (MEF) cell line model to screen for drugs with activity against KRAS mutant tumors revealed a potentially new approach. Lai et al. generated isogenic MEF cell lines with either KRAS wild-type or KRAS-mutated alleles, and subsequently performed phenotypic screening of a small molecule library to discover clusters of compounds with similar mechanisms of action that all shared the feature of being potent inhibitors of KRAS-mutated MEFs. Kinome profiling of the compounds of interest demonstrated one cluster to share the property of being cyclin-dependent kinase 9 (CKD9) inhibitors. CDK9 is a transcriptional co-activator and promotes RNA Polymerase II transition from the promoter-paused state to the elongation state of transcription. CDK9 inhibitory activity was experimentally confirmed using LS513 (KRAS G12D), SW620 (KRAS G12V), and Colo320 (RAS WT) tumor cell line models, and demonstrated the effectiveness of this group of molecules in treating cancer harboring KRAS mutations [121]. This study revealed promising anti-tumor activity by CDK9-targeting compounds as a potential therapeutic strategy for KRAS-mutant CRC. While newer, specific CDK9 inhibitors are undergoing early-phase clinical trial testing, the concept of specifically targeting KRAS-mutated CRC with CDK9 inhibitors has not yet been tested clinically.

## 5. Conclusions

For many years, KRAS mutations were considered to be “undruggable” due to the challenges of targeting the protein’s structure. This has hindered the discovery of new drugs for a large fraction of CRCs driven by this gene mutation. Fortunately, with persistent research efforts, there has been a recent paradigm shift in the field of KRAS drug development, with multiple proof-of-concept approaches leading to objective tumor responses and drug approvals. Initial approaches to KRAS inhibition targeted the G12C mutant form of the protein and relied on small-molecule inhibitors. However, new approaches are now being studied, including novel combinations, cellular therapies, and vaccines with broader KRAS targets. The future is now unexpectedly promising for the ever-ubiquitous, but ever-elusive target that is KRAS. We are only just beginning to understand the full potential and scope of its modulation.

## Figures and Tables

**Figure 1 ijms-24-12030-f001:**
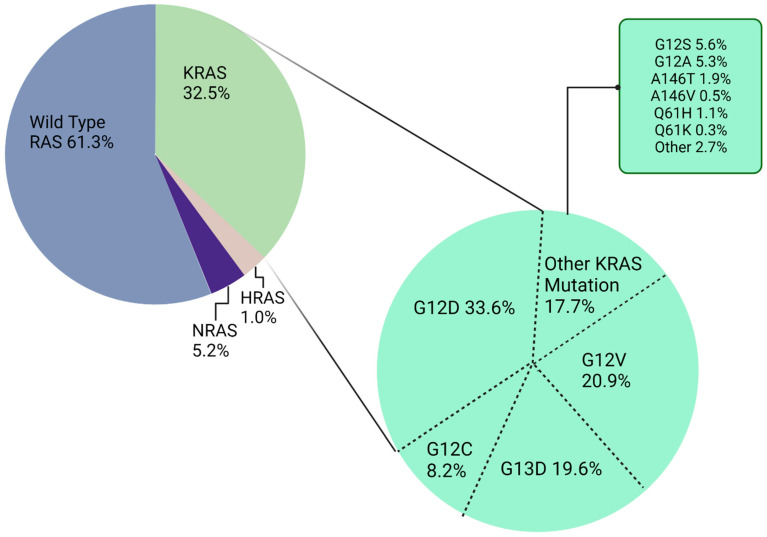
Percentage of colorectal cancers found to harbor different *RAS* mutations.

**Figure 2 ijms-24-12030-f002:**
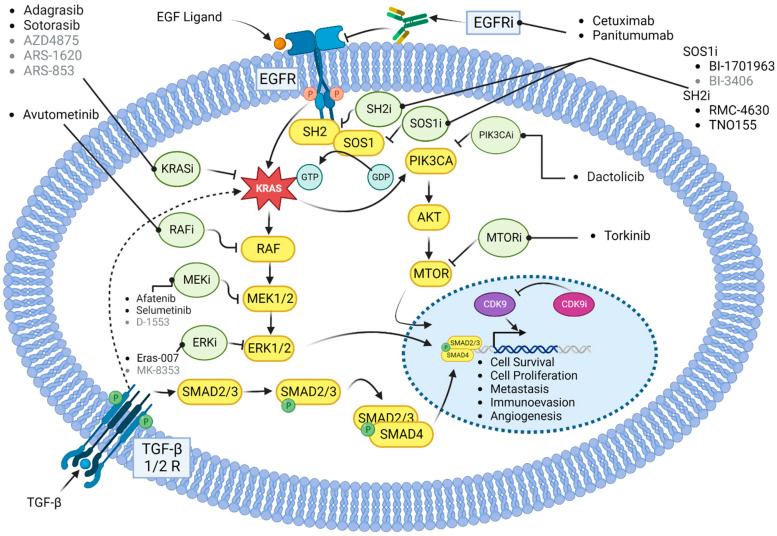
Signaling pathways with drug targets and associated drugs that are described in the text. Bold-lettered drugs are in clinical trials or clinically approved, while gray-lettered drugs have not yet entered clinical investigation. Yellow circles represent proteins in the signaling pathways. Green circles represent classes of drugs designed to target the signaling proteins.

## Data Availability

Not applicable.

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
