# Peer review of "Targeting KRAS in Colorectal Cancer: A Bench to Bedside Review"

_ijms, 2023, doi:10.3390/ijms241512030_

Round 1

Reviewer 1 Report

The paper "Targeting KRAS in colorectal cancer: A bench to bedside review" by Bteich and colleagues is a review about the KRAS gene role in the cancer of colon and rectum (CRC). The review is well written, covering with sufficient mastery the aspects of genetics, molecular biology, pharmacology and clinics, and ultimately providing a complete and interesting perspective on the role of this oncogene in colorectal cancer.

- Figure 1: would it be possible to add a few more mutations to the "Other" panel, beyond the G12A at 5.3%? It is important to notice that not only mutations in G12 and G13 have a pro-oncogenic role, but also, for example, Q61 or A146.

- Would it be possible to comment on the subject of drug repurposing? Paragraphs 3 and 4, about the inhibition of KRAS, could have and extended impact if such strategies could be migrated to other tumors with KRAS mutations as drivers (like lung adenocarcinoma and pancreas cancer).

- Just as a matter of structure, I believe having 17 paragraphs is a bit odd. The review is currently structured in 3 parts: 1) introduction (also about KRAS genetics and function) 2) current strategies for KRAS inhibition 3) future directions. The authors could leave the current paragraphs as subsections of the three main sections. This would help people that are already knowledgeable on KRAS role or current strategies, and would like to skip to the most updated parts of the review.

- Figure 2: the cascade following KRAS activation in the nucleus has also other consequences beyond cell survival, proliferation and metastasis. One being immunoevasion, another one being angiogenesis. In other words, KRAS is a devastating pan-activator of the majority of hallmarks of cancer.

Author Response

Dear Reviewer,

We appreciate your taking the time to help improve our manuscript.  We have made the following revisions.

- Figure 1: would it be possible to add a few more mutations to the "Other" panel, beyond the G12A at 5.3%? It is important to notice that not only mutations in G12 and G13 have a pro-oncogenic role, but also, for example, Q61 or A146.

Response: We agree that there are other notable oncogenic KRAS mutations. We have included these in the revised Figure 1.

- Would it be possible to comment on the subject of drug repurposing? Paragraphs 3 and 4, about the inhibition of KRAS, could have and extended impact if such strategies could be migrated to other tumors with KRAS mutations as drivers (like lung adenocarcinoma and pancreas cancer).

Response: We agree that drug repurposing is a powerful method to discover new treatments for KRAS mutated cancers. We have added a new paragraph discussing drug repurposing.

- Just as a matter of structure, I believe having 17 paragraphs is a bit odd. The review is currently structured in 3 parts: 1) introduction (also about KRAS genetics and function) 2) current strategies for KRAS inhibition 3) future directions. The authors could leave the current paragraphs as subsections of the three main sections. This would help people that are already knowledgeable on KRAS role or current strategies, and would like to skip to the most updated parts of the review.

Response: We agree that the organization of the manuscript can be improved. We have revised to make the 3 major sections as suggested.

  • Figure 2: the cascade following KRAS activation in the nucleus has also other consequences beyond cell survival, proliferation and metastasis. One being immunoevasion, another one being angiogenesis. In other words, KRAS is a devastating pan-activator of the majority of hallmarks of cancer.

Response: We agree that these other oncogenic properties of KRAS mutation should be considered when thinking about KRAS biology and targeting KRAS mutated cancers. We have included these consequences in the revised Figure 2.

Reviewer 2 Report

In this review, the authors present an overview of targeting KRAS in colorectal cancer.  Considering the presence of KRAS mutations in colorectal cancer and how KRAS mutations are difficult to treat, there is an urgent need to address the unmet medical need by using monotherapy or combinational targeted therapies.

In this review they have focused mostly on targeting KRAS G12C but have not talked about KRAS G12D inhibitor (MRTX1133) so studies of KRAS G12D inhibitor in CRC can be considered.

1. The authors could discuss the ongoing study of small molecule KRAS inhibitor MRTX1133 in CRC as  there is ongoing clinical trials.

2. The authors can also discuss the combination therapies for MRTX1133 as it is known that KRAS inhibition by targeted cancer therapies is known to upregulate RTK which can inhibited by combination mechanism.

The review have covered most of the targeted cancer therapy studies for KRAS G12C and including the MRTX1133 (KRAS G12D inhibitor) studies can improve the manuscript.

Author Response

Dear Reviewer,

We appreciate your taking the time to review our manuscript and help improve it. We have revised the manuscript, with the following changes in response to your suggestions.

  1. The authors could discuss the ongoing study of small molecule KRAS inhibitor MRTX1133 in CRC as there is ongoing clinical trials.

Response: We agree that this was a key omission in the original version. Our revision has a new paragraph discussing MRTX1133 preclinical and clinical studies.

  1. The authors can also discuss the combination therapies for MRTX1133 as it is known that KRAS inhibition by targeted cancer therapies is known to upregulate RTK which can inhibited by combination mechanism.

Response: Our MRTX1133 paragraph also includes discussion about combination therapies for MRTX1133.